# Effect of the Intake of Solid Block Dairy Products Like Cheese on Serum Uric Acid in Children: A Preliminary Mechanistic Investigation

**DOI:** 10.3390/nu16223864

**Published:** 2024-11-12

**Authors:** Zhongting Lu, Zhenchuang Tang, Xin Guo, Lei Liu, Xuemei Cheng, Lianlong Yu, Guangyan Cheng

**Affiliations:** 1School of Public Health, Cheeloo College of Medicine, Shandong University, Jinan 250012, China; 202216405@mail.sdu.edu.cn (Z.L.); xguo@sdu.edu.cn (X.G.); 2Institute of Food and Nutrition Development, Ministry of Agriculture and Rural Affairs, Beijing 100081, China; tangzhenchuang@caas.cn; 3Shandong Center for Disease Control and Prevention, Jinan 250014, China; 18615281821@163.com (L.L.); meixuecheng@126.com (X.C.)

**Keywords:** solid block dairy products, cheese, serum uric acid levels, children and adolescents, Mendelian randomization

## Abstract

Objective: This study aimed to investigate the relationship between the intake of solid block dairy products like cheese and serum uric acid levels, along with its potential physiological mechanisms. Methods: Data for our study were obtained from the Chinese Children and Lactating Women Nutrition and Health Surveillance. Generalized linear models and restricted cubic splines were employed to analyze the relationship between the intake of solid block dairy products like cheese and serum uric acid levels. Two-sample Mendelian randomization (TSMR) analysis was conducted to infer causality, based on a large sample size and robust methodology. Gene Ontology (GO) enrichment analysis was also performed to identify potential biological pathways. Results: Among all types of dairy products, a significant negative association with serum uric acid levels was observed only for the intake of solid block dairy products like cheese, regardless of covariate adjustment (β = −0.182, *p* < 0.001). TSMR results supported a negative causal relationship between cheese intake and serum uric acid levels (β = −0.103, 95% CI: −0.149 to −0.057; *p* = 0.002). The JAK-STAT signaling pathway and autophagy regulation were identified as potential physiological mechanisms underlying this relationship. Conclusions: The intake of solid block dairy products like cheese was found to result in decreased levels of serum uric acid, with potential mechanisms involving the JAK-STAT signaling pathway and the regulation of autophagy.

## 1. Introduction

Hyperuricemia is a metabolically disordered condition with the presence of abnormally elevated serum uric acid levels [1]. Hyperuricemia has increasingly become a significant public health issue worldwide [2]. The incidence of Hyperuricemia has remarkably increased by about 20% worldwide [3]. In China, the prevalence of hyperuricemia was estimated at 16.4% [4], with a notable rise in prevalence among younger populations in recent years [5,6]. Uric acid, a metabolite resulting from purine metabolism, is closely associated with renal excretion function, dietary habits, and genetic factors [7]. Elevated serum uric acid levels have been substantially linked to increased risks of gout, hypertension, metabolic syndrome, chronic kidney disease, and cardiovascular diseases [8,9]. Thus, exploring effective dietary interventions to lower serum uric acid levels is of considerable public health importance.

With improved living standards and greater dissemination of nutritional knowledge, dairy product consumption has progressively increased. Components within dairy products, such as calcium, casein, and whey protein, have been suggested to play a role in regulating uric acid metabolism [10,11]. Cheese, a concentrated and fermented dairy product, is renowned for its high nutritional value. It is characterized by its high-fat content, smooth texture, and richness in high-quality proteins, calcium, phosphorus, and various vitamins, earning it the designation of “the golden milk”. Cheese is composed of several key ingredients, the primary element in cheese production remains milk (80–90%), rennet (0.05–0.1%), bacterial cultures (0.5–1%), salt (1–2%), and flavorings [12].

Previous studies have investigated the effects of dairy products, such as milk and yogurt, on serum uric acid levels in adults and identified certain uric acid-lowering benefits [13,14]. High consumption of total dairy products, total milk, low-fat dairy products, low-fat milk, low-fat yogurt, and cheese were associated with a lower risk of hyperuricemia [15]. However, research on children and adolescents, especially regarding concentrated dairy products like cheese, remained limited. Childhood and adolescence are critical periods for growth and development and for establishing dietary habits. This study hypothesized that there was a significant correlation between cheese intake and serum uric acid levels in Chinese children and adolescents. Therefore, this study aimed to investigate the relationship between cheese consumption and serum uric acid levels in Chinese children and adolescents, addressing the gap in studies focused on this age group. Furthermore, we also employed a two-sample Mendelian randomization (TSMR) approach to infer the causal association between these variables and explored potential mechanisms at the level of mapped genes and biological pathways using single nucleotide polymorphisms (SNPs).

## 2. Materials and Methods

### 2.1. Study Participants

Data for this study were obtained from the Chinese Children and Lactating Women Nutrition and Health Surveillance (CCLWNHS), conducted from 2016 to 2019. This dataset encompassed comprehensive basic, nutritional, and health information for all participants. A multi-stage stratified cluster sampling method was employed, selecting 26 primary schools, 26 junior high schools, and 13 high schools across 13 districts and counties in Shandong Province. The total sample included 3568 participants aged 6–18. Ethical approval for the study was granted by the Ethics Committee of the National Institute for Nutrition and Health and the Chinese Center for Disease Control and Prevention (approval number: 201614, approval date: 3 June 2016). All participants provided written informed consent and voluntarily engaged in the survey.

### 2.2. Outcome Variables

Fasting venous blood samples (6 mL) were collected from all participating children and adolescents. The serum was separated and stored at −80 °C before being transported via a low-temperature chain to the central laboratory of the Chinese Center for Disease Control and Prevention for standardized quantitative analysis. The following biochemical and nutritional indicators were measured: serum uric acid (Cat#: 05185121160, Roche Diagnostics, Jinan, China) (CV ≤ 5.0%), fasting blood glucose (FBG) (Cat#: 05185125160, Roche Diagnostics Jinan, China) (CV ≤ 5.0%), high-density lipoprotein cholesterol (HDL-C) (Cat#: 05185149160, Roche Diagnostics, Jinan, China) (CV ≤ 5.0%), low-density lipoprotein cholesterol (LDL-C) (Cat#: 05185151160, Roche Diagnostics, Jinan, China) (CV ≤ 5.0%), triglycerides (TG) (Cat#: 05185140160, Roche Diagnostics, Jinan, China) (CV ≤ 5.0%), and total cholesterol (TC) (Cat#: 05185139160, Roche Diagnostics, Jinan, China) (CV ≤ 5.0%). The above biochemical indicators were detected by Roche Cobas c702 (Cat#: 06473245001, Roche Diagnostics, Jinan, China).

### 2.3. Exposure Variable

A standardized food frequency questionnaire incorporating a personal nutrition and health survey was used to collect data on the consumption patterns, frequencies, and quantities of various foods over the past 30 days for children and adolescents aged 6–18. Trained interviewers conducted face-to-face interviews in participants’ homes, ensuring data quality by excluding any substandard responses. The final dataset related to dairy product intake included information on the intake of solid block dairy products such as cheese (g/day), liquid milk (g/day), powdered milk (g/day), and yogurt (g/day).

### 2.4. Covariates

Various potential confounding factors were considered in the analysis. Demographic covariates included age and gender (“male” and “female”). Socioeconomic factors encompassed place of residence (“rural” and “ urban”), the number of days per week exposed to secondhand smoke (defined as smoke exhaled by smokers or emitted from the burning end of cigarettes), and alcohol consumption within the past 30 days (“consumed alcohol within a month”, “consumed alcohol within the past month”, and “never”). Health behavior variables included body mass index (BMI), energy intake, and the amount of moderate to vigorous physical activity.

### 2.5. Statistical Analysis

Differences between variables for males and females were assessed using Student’s *t*-tests and chi-square tests. The effects of different dairy product intake on serum uric acid levels were analyzed using generalized linear models (GLMs), with age, gender, BMI, residence, physical activity, energy intake, exposure to secondhand smoke, and alcohol consumption adjusted. All effects were reported as β coefficients with corresponding 95% confidence intervals (CIs). Sensitivity analysis was conducted to ensure the results were robust, including stratified and interaction analysis. The samples were stratified into different subgroups according to different age groups based on the average age of 11, gender, residence, and so on. Interaction effects were examined by including interaction terms in separate models. Restricted cubic splines (RCS) were employed to analyze the dose–response relationship between the intake of solid block dairy products like cheese and serum uric acid levels. The analysis was performed using R (version 4.4.0). Statistical significance was defined as a *p*-value of <0.05.

### 2.6. TSMR Analysis

This study employed the IEU Open GWAS database (https://gwas.mrcieu.ac.uk/datasets, accessed on 27 October 2024) and applied a TSMR approach to investigate the causal relationship between cheese intake and serum uric acid levels, since the negative association was only observed with the intake of solid block dairy products like cheese among all dairy items. Cheese intake (GWAS ID: ukb-b-1489) was used as the exposure variable, while serum uric acid levels (GWAS ID: ebi-a-GCST90018977) served as the outcome variable. Inverse Variance Weighting (IVW) served as the primary analytical method. Statistical analysis and result visualizations were performed using the R packages ‘TwoSampleMR’ and ‘ieugwasr’ (version 4.4.0).

### 2.7. Bioinformatics Analysis

For bioinformatics analysis, the Functional Mapping and Annotation of Genome-Wide Association Studies (FUMA) platform was utilized to annotate, visualize, and interpret results from SNPs (https://fuma.ctglab.nl, accessed on 27 October 2024) [16]. FUMA analysis contains two core functions: SNP2GENE and GENE2FUNC. The SNP2GENE function takes GWAS summary statistics as an input and provides extensive functional annotation for all SNPs in genomic areas identified by lead SNPs. The GENE2FUNC function takes a list of gene IDs and annotates genes in a biological context. In the SNP2GENE process, the independent significant SNPs were defined as SNPs with *p* < 5 × 10^−6^ and independence at r^2^ < 0.6 within 1 Mb, among which the lead SNPs are still independent when r^2^ < 0.1. Genomic risk loci were identified by merging the LD blocks of independent significant SNPs close to each other within 200 kb. Single nucleotide polymorphism-based tissue enrichment analysis, facilitated by FUMA, was employed to scrutinize phenotype-related tissue specificity. Based on the GTEx v6 RNA-seq data, tissue-specific expression patterns for each gene are visualized as an interactive heatmap. This method utilizes gene-property analyses to investigate the correlation between GWAS hits and tissue-specific gene expression profiles. Genes mapped within a 200-base pair region flanking each SNP were identified using the R package ‘vautils’ (version 4.4.0). Following this, Gene Ontology (GO) enrichment analysis was conducted using the DAVID database (https://david.ncifcrf.gov, accessed on 27 October 2024). Furthermore, a protein–protein interaction (PPI) network was constructed with STRING version 12.0 (https://cn.string-db.org, accessed on 27 October 2024) and visualized using Cytoscape (version 9.3.0).

## 3. Results

### 3.1. Descriptive Statistics

The study included 3568 children and adolescents from CCLWNHS, following the exclusion of samples with insufficient data quality. Table 1 outlines the fundamental characteristics of the participants. The mean age of the cohort was 11.38 years (Standard Deviation = 3.16), with a nearly even distribution of 1775 males (49.75%) and 1793 females (50.25%). The average BMI of female participants was 18.94 kg/m^2^, which was significantly lower than the 19.82 kg/m^2^ observed in male participants. There were no significant differences between the genders in the intake of solid block dairy products like cheese, liquid milk, powdered milk, yogurt, protein, or fat. However, female participants had lower carbohydrate and energy intakes compared to their male counterparts (*p* < 0.05). The mean serum uric acid level for male participants was 338.75 µmol/L (Standard Deviation = 90.25), whereas female participants had a lower mean serum uric acid level of 294.49 µmol/L (Standard Deviation = 67.20).

### 3.2. Association Analysis of Diverse Dairy Products and Serum Uric Acid Levels

Table 2 details the associations between various dairy products and serum uric acid levels. After adjusting for covariates, only the intake of solid block dairy products, such as cheese, showed a consistently significant negative association with serum uric acid across all models. In Model 1, the association was β = −0.098 (95% CI: −0.192, −0.003), and in Model 2, it was β = −0.244 (95% CI: −0.244, −0.081). Model 3, which included full covariate adjustment, revealed increased effect estimates and confidence intervals compared to Models 1 and 2 (β = −0.182, 95% CI: −0.269, −0.095), with all results remaining statistically significant.

The stratified and interaction analysis conducted in Model 3 was summarized in Table 3. In both genders, the intake of solid block dairy products like cheese was significantly negatively associated with serum uric acid levels (β = −0.179, 95% CI: −0.307 to −0.052 for males; β = −0.227, 95% CI: −0.339 to −0.115 for females). Among individuals aged 11 years or younger, the effect of cheese intake on serum uric acid levels was not significant (*p* = 0.158). The negative association between cheese intake and serum uric acid levels was more pronounced in participants with lower BMI compared to those with overweight or obesity (β = −0.134, 95% CI: −0.248 to −0.020). This negative association was also significant in participants with higher daily energy intake (β = −0.178, 95% CI: −0.263 to −0.092). Furthermore, the relationship between cheese intake and serum uric acid levels did not vary by urban or rural residence. None of the interactions between these variables and the intake of solid block dairy products like cheese were statistically significant (*p*_interaction > 0.05).

The dose–response relationship between the intake of solid block dairy products like cheese and serum uric acid levels was examined using restricted cubic spline (RCS) analysis. After adjusting for all covariates, it is more intuitively observed that the level of serum uric acid decreased with the intake of solid block dairy products like cheese (Figure 1).

### 3.3. Two-Sample Mendelian Randomization Analysis of PM2.5 and Sodium in Urine

The IVW method revealed a significant negative causal link between cheese consumption and serum uric acid levels (β = −0.103, 95% CI: −0.149 to −0.057; *p* = 0.002). Similarly, the Weighted Median method yielded comparable results (β = −0.130, 95% CI: −0.159 to −0.101; *p* < 0.001) (Figure 2A,B). The adjusted Q and I2 statistics indicated the presence of heterogeneity (*p* < 0.05), which was further explored using MR-Presso. Subsequent MR-Egger tests on the selected SNPs did not reveal significant evidence of potential horizontal pleiotropy (*p* = 0.851). Sensitivity analysis was performed using the leave-one-out method. Confidence intervals for all SNPs were consistently positioned to the left of zero (Figure 2C), indicating that the exclusion of individual SNPs did not substantially alter the results.

Positional and expression analysis of SNPs was performed using FUMA. These SNPs were precisely mapped to 206 nearest genes, and their expression levels across various tissues were visualized in the heatmap (Appendix A). Then, GO enrichment analysis indicated that the mapped genes were significantly enriched in biological pathways, such as the JAK-STAT signaling pathway, growth and development, synaptic regulation, nutrient metabolism, and autophagy (Figure 3A). The analysis of PPI networks analysis highlighted that proteins such as E1A binding protein P300 (EP300), microtubule-associated protein tau (MAPT), and nuclear protein 1 (NUPR1), which are implicated in autophagy regulation, occupy central positions with high degree centrality. Additionally, chorionic somatomammotropin hormone 1 (CSH1), chorionic somatomammotropin hormone 2 (CSH2), growth hormone 1 (GH1), and growth hormone 2 (GH2), which are associated with growth and development, as well as components of the JAK-STAT signaling pathway, also exhibit relatively high degree centrality (Figure 3B).

## 4. Discussion

This study established an epidemiological validation of a significant inverse relationship between the consumption of solid block dairy products, such as cheese, and serum uric acid levels among Chinese children and adolescents (β = −0.182, *p* < 0.001). This association remained statistically significant after adjusting for various covariates. Additionally, Mendelian randomization analysis confirmed a causal negative relationship between cheese intake and serum uric acid levels. Investigations at the single nucleotide polymorphism (SNP), gene, and biological pathway levels suggested that potential physiological mechanisms underlying this association may involve the JAK-STAT signaling pathway and the regulation of autophagy.

Excessive uric acid production and impaired uric acid excretion were identified as key mechanisms underlying hyperuricemia [18]. The regulation of uric acid homeostasis thus emerged as a crucial aspect of hyperuricemia and gout treatment [19]. Excessive uric acid production was primarily attributed to genetic mutations or functional abnormalities in enzymes related to purine metabolism, high-purine diets, fructose intake, and alcohol consumption, all of which directly or indirectly influence purine metabolism and contribute to uric acid overproduction [20,21]. Conversely, impaired uric acid excretion was associated with alterations in transport proteins (e.g., URAT1, OAT4, and OAT10) involved in uric acid excretion and reabsorption in the renal tubules, leading to compromised urinary uric acid excretion [22,23].

In recent years, the relationship between dairy products and serum uric acid levels has garnered significant attention. Multiple studies have indicated an association between dairy consumption and lower uric acid levels [11,24]. Analysis from the Third National Health and Nutrition Examination Survey (NHANES III) demonstrated an inverse relationship between dairy intake and serum urate concentrations, with individuals consuming milk at least once daily exhibiting lower serum urate levels compared to non-consumers (multivariate difference, −0.25 mg/dL [95% CI, −0.40 to −0.09]) [24]. Liquid milk, primarily composed of proteins (approximately 3.2% casein and whey proteins), fats (about 3.7%), lactose (around 4.6%), and various vitamins and minerals [25], has been shown to impact uric acid levels. A randomized controlled crossover study further revealed that casein or whey protein could enhance uric acid clearance, significantly reducing serum urate levels within 3 h [26].

Cheese, a concentrated solid dairy product, is rich in high-quality proteins (mainly casein), lipids, minerals (such as calcium, phosphorus, and magnesium), and vitamins (including vitamins A, K2, B2, B12, and folate), as well as probiotics and bioactive molecules (e.g., bioactive peptides, lactoferrin, short-chain fatty acids, and milk fat globule membrane). Consumption of approximately 40 g/day of cheese has been associated with various health benefits, including a reduced risk of metabolic and cardiovascular diseases [12,27,28]. In our study, a significant negative association between cheese intake and serum uric acid levels was observed, with this inverse relationship being particularly pronounced in children and adolescents over 11 years of age who were not overweight or obese and independent of gender (Table 3). This may be attributed to the higher levels of uric acid-regulating active components, such as casein and whey proteins, present in cheese. The synergistic effects of these components might represent a key mechanism for lowering serum uric acid levels. Conversely, intake of liquid milk, yogurt, and milk powder did not demonstrate a statistically significant association with serum uric acid levels (Table 2).

Mendelian randomization analysis served as a supplementary method to infer the causal relationship between cheese consumption and serum uric acid concentration, addressing the limitations inherent in cross-sectional studies. The results from the Inverse Variance Weighted (IVW) analysis revealed a significant negative causal relationship between cheese intake and serum uric acid levels (β = −0.103, 95% CI: −0.149 to −0.057; *p* = 0.002). The TSMR findings of this study were deemed robust and reliable, suggesting that cheese consumption could contribute to a reduction in serum uric acid concentrations and may act as a protective factor.

To explore the potential mechanisms underlying this effect, we examined the functional effect SNPs identified, which localized to 206 nearest genes. Generally, these nearest genes are considered the most probable targets influenced by the SNPs. GO biological pathway enrichment analysis and PPI network analysis of these genes indicated enrichment in pathways related to growth and development, the JAK-STAT signaling pathway, autophagy, and synaptic regulation. The JAK-STAT (Janus kinase-signal transducer and activator of transcription) signaling pathway is a crucial intracellular signaling system involved in various physiological and pathological processes, including immune regulation, cell proliferation, and differentiation [29,30,31]. Given that uric acid salts can directly or indirectly activate various immune cells, leading to the production of inflammatory cytokines [32], which may impact the kidneys and subsequently affect uric acid excretion [33], this pathway’s role in inflammation and metabolic syndrome, both associated with uric acid levels, is particularly noteworthy [34,35]. Autophagy, an intracellular “clean-up” mechanism that maintains cellular homeostasis by degrading damaged organelles and misfolded proteins via lysosomal pathways [36,37,38], also plays a role in this context. The NLRP3 inflammasome, which recognizes uric acid salts, is involved in the inflammatory response associated with hyperuricemia [39,40]. Autophagy interacts with the NLRP3 inflammasome [41,42], and prolonged renal tubular autophagy could potentially lead to irreversible cellular damage or loss of uric acid transporter function, thereby affecting uric acid metabolism and excretion [43]. Although direct studies linking the JAK-STAT signaling pathway with uric acid metabolism are limited, the existing evidence suggests its involvement in inflammation and metabolic syndrome. It was suggested that future animal experiments be conducted to further explore the specific mechanisms of the JAK-STAT signaling pathway and autophagy in uric acid metabolism. These relative experimental studies could clarify whether the physiological connection between cheese intake and the reduction in serum uric acid levels involved the JAK-STAT signaling pathway and autophagy.

This study demonstrated significant strengths by focusing on a substantial cohort of children and adolescents and employing both GLM and Mendelian Randomization analysis. These methodological approaches provided robust evidence supporting the association between cheese consumption and serum uric acid levels, thereby enhancing the reliability of the findings. This evidence also offers a theoretical foundation for developing dietary intervention strategies targeted at children and adolescents, particularly in the context of the rising incidence of hyperuricemia within younger populations. However, there are notable limitations to consider. Hyperuricemia includes a specific genetic background with a familiar history. However, the genetic factors of hyperuricemia in the participants could not be analyzed in this study, since hyperuricemia in the family history section of the questionnaire was not obtained. The fat content of the dairy products and other interfering dietary habits were not acquired. The incidence of basic diseases among them is very low, with little or no medicine use recorded. Given that accurate information on the type, dosage, and duration of medicine use was difficult to obtain and verify, the inclusion of medicine use information may bring about complexity and uncertainty. Our current study, with its limited sample size and scope, as well as, does not provide sufficient data to support the derivation of a definitive threshold with statistically significant and clinically relevant. The potential for other adverse effects associated with excessive cheese consumption necessitates a multifaceted assessment. A cheese intake threshold remains to be explored. The restricted sample size of the Asian dataset and the reliance on Mendelian Randomization analysis based on European populations may limit the generalizability of the results to Asian populations. Furthermore, while bioinformatics analysis has provided valuable insights, the specific mechanisms through which cheese intake influences serum uric acid levels remain to be experimentally validated. Further research is necessary to elucidate these mechanisms and to confirm the applicability of these findings across diverse populations.

## 5. Conclusions

Serum uric acid levels showed significant negative correlations with the intake of solid block dairy products like cheese in Chinese children and adolescents. The intake of solid block dairy products resulted in decreased serum uric acid levels, potentially involving the JAK-STAT signaling pathway and the regulation of autophagy as underlying physiological mechanisms.

## Figures and Tables

**Figure 1 nutrients-16-03864-f001:**
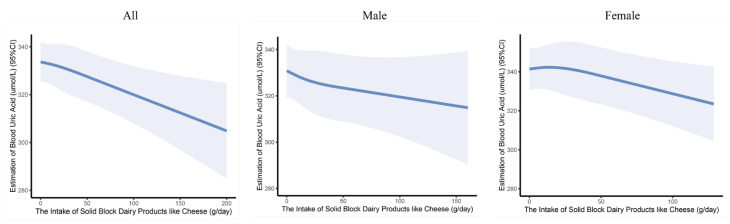
The dose–response curves of the intake of solid block dairy products like cheese and serum uric acid levels in children and adolescents. The associations were adjusted for age, gender, BMI, residence, physical activity time, energy intake, exposure to secondhand smoke, and alcohol consumption.

**Figure 2 nutrients-16-03864-f002:**
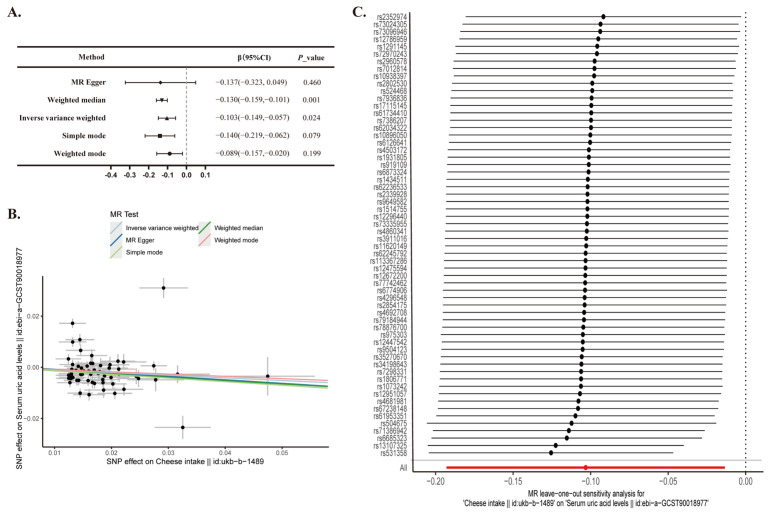
Two-sample Mendelian randomization analysis of cheese intake and serum uric acid levels. (**A**) Mendelian Randomization; CI, confidence interval. (**B**) A scatter plot is used to visually represent the causal relationship between cheese intake and serum uric acid levels. (**C**) A forest plot depicting the ‘leave-one-out’ sensitivity analysis method to illustrate the impact of individual SNPs on the outcomes.

**Figure 3 nutrients-16-03864-f003:**
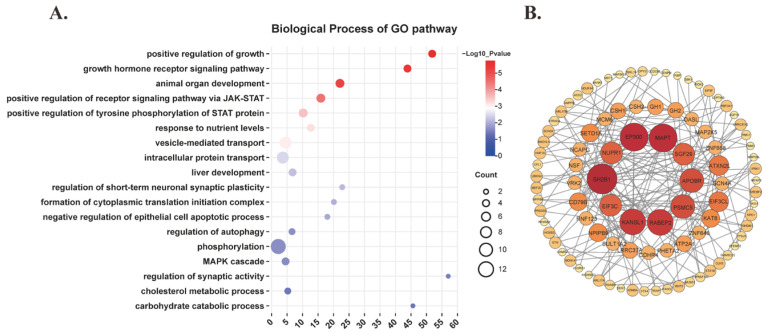
Bioinformatics analysis of genes mapped by functional SNPs. (**A**) Biological process of GO pathway. (**B**) PPI network.

**Table 1 nutrients-16-03864-t001:** Basic characteristics of children and adolescents in China.

Variables	Total	Male	Female	*p* Value
Gender, *n* (%)	3568	1775 (49.75)	1793 (50.25)	
Age (years)	11.38 ± 3.16	11.42 ± 3.15	11.34 ± 3.16	0.449
Height (cm)	150.52 ± 16.70	152.60 ± 18.20	148.46 ± 14.80	<0.001
Weight (kg)	45.13 ± 16.55	47.58 ± 18.27	42.71 ± 14.24	<0.001
BMI (kg/m^2^)	19.38 ± 5.87	19.82 ± 6.07	18.94 ± 5.63	<0.001
Residence [17]				0.899
Rural, n (%)	1936 (54.26)	965 (54.37)	971 (54.16)	
Urban, n (%)	1632 (45.74)	810 (45.63)	822 (45.84)	
Solid block dairy products consumption (g/day)	7.03 ± 28.77	7.58 ± 29.43	6.50 ± 28.10	0.261
Liquid milk consumption (g/day)	77.96 ± 118.99	75.24 ± 130.50	80.66 ± 106.34	0.174
Powder milk consumption (g/day)	1.80 ± 15.21	1.71 ± 16.29	1.90 ± 14.07	0.708
Yogurt consumption (g/day)	55.18 ± 92.28	57.73 ± 82.09	52.66 ± 101.68	0.102
Protein intake (g/day)	139.81 ± 131.59	143.42 ± 150.25	136.24 ± 109.97	0.104
Fat intake (g/day)	52.15 ± 76.18	52.29 ± 71.11	52.00 ± 80.91	0.911
Carbohydrate intake (g/day)	393.96 ± 335.17	409.88 ± 361.78	378.21 ± 305.85	0.005
Energy intake (kcal/day)	2391.81 ± 2102.69	2466.79 ± 2140.31	2317.59 ± 2062.69	0.034
Physical Activity (min)	39.41 ± 46.60	42.42 ± 48.65	36.42 ± 44.29	<0.001
Exposure days to secondhand smoke per week (days)	4.33 ± 1.19	4.26 ± 1.26	4.40 ± 1.11	<0.001
Alcohol consumption				<0.001
Consumed alcohol within a month	105 (2.94)	85 (4.79)	20 (1.12)	
Consumed alcohol within the past month	170 (4.76)	109 (6.14)	61 (3.40)	
Never	3292 (92.26)	1581 (89.07)	1711 (95.43)	
Serum Uric Acid (mmol/L)	316.51 ± 82.52	338.75 ± 90.25	294.49 ± 67.20	<0.001

Mean ± Standard Deviation. Abbreviations: CI, confidence interval.

**Table 2 nutrients-16-03864-t002:** Multivariable associations between diverse dairy products and serum uric acid levels in children and adolescents.

	Model 1			Model 2			Model 3		
	Coefficient	95% CI	*p* Value	Coefficient	95% CI	*p* Value	Coefficient	95% CI	*p* Value
The intake of solid block dairy products like cheese (g/day)	−0.098	−0.192, −0.003	0.042	−0.163	−0.244, −0.081	<0.001	−0.182	−0.269, −0.095	<0.001
The intake of liquid milk (g/day)	−0.005	−0.028, 0.017	0.649	−0.001	−0.021, 0.019	0.939	0.002	−0.019, 0.022	0.860
The intake of powdered milk (g/day)	−0.008	−0.186, 0.170	0.927	−0.022	−0.176, 0.133	0.783	−0.002	−0.159, 0.155	0.978
The intake of yogurt (g/day)	0.028	−0.001, 0.057	0.060	0.015	−0.010, 0.041	0.235	0.027	−0.001, 0.054	0.058

Note: Abbreviations: CI, confidence interval; Model 1 unadjusted. Model 2 adjusted for age, gender, and BMI. Model 3 adjusted for age, gender, BMI, residence, physical activity time, energy intake, exposure to secondhand smoke, and alcohol consumption.

**Table 3 nutrients-16-03864-t003:** Multivariate stratified analyses of the association between solid block dairy product consumption and serum uric acid levels.

	N (%)	Coefficient	95% Cl	*p* Value	*p* for Interaction
Gender					0.235
Male	1775 (49.75)	−0.179	−0.307, −0.052	0.006	
Female	1793 (50.25)	−0.227	−0.339, −0.115	<0.001	
Age					0.987
≤11	1874 (52.52)	−0.113	−0.270, 0.044	0.158	
>11	1694 (47.48)	−0.222	−0.330, −0.113	<0.001	
Overweight or obese					0.267
No	2637 (73.93)	−0.134	−0.248, −0.020	0.022	
Yes	930 (26.07)	−0.014	−0.179, 0.152	0.869	
Residence					0.545
Rural	1936 (54.26)	−0.225	−0.443, −0.007	0.043	
Urban	1632 (45.74)	−0.174	−0.271, −0.077	<0.001	
Energy intake					0.827
≤P50	1784 (50.00)	0.177	−0.191, 0.545	0.346	
>P50	1784 (50.00)	−0.178	−0.263, −0.092	<0.001	

## Data Availability

Restrictions apply to the availability of these data. Data were obtained from our project group and are available (https://www.chinacdc.cn, accessed on 27 October 2024) with the permission of our project group.

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
