# Peer review of "Effect of the Intake of Solid Block Dairy Products Like Cheese on Serum Uric Acid in Children: A Preliminary Mechanistic Investigation"

_nutrients, 2024, doi:10.3390/nu16223864_

Round 1
Reviewer 1 Report
Comments and Suggestions for Authors
This is my review on the manuscript titled "Effect of the intake of solid block dairy products like cheese on serum uric acid in Children: Causal Investigation and Mechanistic Exploration".
The introduction is generally fine. The research hypothesis should be stated more clearly.
Methods are well described. You should include a further analysis of sample size, especially aiming to compare subgroups (e.g., different age groups, genders, or BMI categories) and on other confounders that may influence uric acid levels!
Statistics are ok.
In the discussion, you should include practical implications for your results. Your discussion should provide specific recommendations for future research.
Regarding the JAK-STAT signaling pathway, suggest for future experimental studies on the matter, to confirm the role of these pathways.
Author Response
- The introduction is generally fine. The research hypothesis should be stated more clearly.
Response: Thank you very much for your suggestions. At your suggestion, the research hypothesis has been stated in introduction according to your suggestion.
- Methods are well described. You should include a further analysis of sample size, especially aiming to compare subgroups (e.g., different age groups, genders, or BMI categories) and on other confounders that may influence uric acid levels!
Response: Thank you very much for your suggestions. We have supplemented the description of the stratified analysis in the section of method and the sample sizes included in the different subgroups in the Table 3. Thank you again for your suggestion.
- Statistics are ok.
Response: Thank you very much for your comments.
- In the discussion, you should include practical implications for your results. Your discussion should provide specific recommendations for future research.
Response: Thank you very much for your suggestions. According to your advice, we have shown practical implications and specific recommendations such as future experimental studies have been provided in discussion. Thank you again for your professional advice.
- Regarding the JAK-STAT signaling pathway, suggest for future experimental studies on the matter, to confirm the role of these pathways.
Response: Thank you very much for your suggestions. Based on your comments, we have suggest relative experimental research to confirm the role of these pathways. Thank you again for your constructive comments.
Reviewer 2 Report
Comments and Suggestions for Authors
This is a comprehensive and ambitious study, which is nevertheless limited by the methodology employed and the highly speculative conclusions and discussions based on the results obtained by the Authors. Here are my main points of criticism:
1. Although English language is comprehensible and generally fine, its should be improved, as there are some minor typo and syntax errors. Before the Conclusions section, the phrase "This section is not mandatory...." should be removed.
2. Although the statistical methods used are powerful, there are many confounding factors that were not accounted for, so some of the results are questionable. For instance, the fat content of the dairy products is not specified, other interfering dietary habits are not documented, potential therapies and long term used of medicines/drugs is not accounted for, etc. All these missing parameters make the statistical results unreliable.
3. The abstract should provide more thorough information about the genomic studies and their results; only JAK pathway is mentioned, without any additional explanation.
4. The genomic study is insufficiently developed. The limited number of genes analyzed and the reduced number of SNPs are insufficient for drawing any solid conclusions. Also, it is well known that epigenetic changes alter lipid and uric acid metabolism, and the Authors mention nothing about this.
5. Did the Authors take any brothers/twins in the study? This should provide some interesting conclusions about the effects of diet in genetically close or similar individuals.
6. The genomic analysis methodology is missing entirely; how was the DNA/RNA prepared, how was the sequencing performed, what controls were used etc.
7. Talking about controls, these are also missing entirely; similar investigations should have been performed on a group of subjects having a non-diary based nutrions, for comparison.
8. The gene expression analysis, SNP analysis and especially the theoretical protein-protein interactions are highly speculative and do not provided reliable information in the absence of mandatory lab experiments.
To conclude, this is an interesting study but the methodology is flawed, the analysis is superficial and the results/conclusions are speculative and questionable.
Author Response
- Although English language is comprehensible and generally fine, its should be improved, as there are some minor typo and syntax errors. Before the Conclusions section, the phrase "This section is not mandatory...." should be removed.
Response: Thank you for your meticulous and professional advice. We appreciate your scrupulous review for our manuscript. And the phrase has been removed. Thank you again for your constructive comments.
- Although the statistical methods used are powerful, there are many confounding factors that were not accounted for, so some of the results are questionable. For instance, the fat content of the dairy products is not specified, other interfering dietary habits are not documented, potential therapies and long term used of medicines/drugs is not accounted for, etc. All these missing parameters make the statistical results unreliable.
Response: Thank you for your valuable feedback on our manuscript. We fully appreciate the concerns raised regarding the potential confounding factors that were not accounted for in our analysis. While we acknowledge that the inclusion of additional variables such as the fat content of dairy products, other dietary habits, potential therapies, and long-term medication use would have enriched our study, we regret to inform you that these data were not collected during the initial survey design due to practical and logistical constraints. Firstly, the survey questionnaire was designed with a focus on capturing key nutritional and health indicators relevant to the primary objectives of the study, which was to assess the association between dairy intake and serum uric acid levels among children and adolescents. The inclusion of detailed dietary fat content, comprehensive dietary habits, and medication use would have significantly increased the complexity and duration of the survey, potentially compromising participation rates and data quality. Secondly, given the scope and resources available for the study, we prioritized the collection of data on dairy intake, demographic characteristics, and other key covariates that were deemed most relevant to our research question. The decision to exclude certain variables was based on a careful balance between the need for comprehensive data and the practical feasibility of conducting the survey. Despite these limitations, we have taken rigorous steps to ensure the robustness of our findings. This includes the use of advanced statistical methods to adjust for the covariates that were available, conducting sensitivity analyses, and carefully interpreting our results in the context of the study design and available data.
We acknowledge that the absence of these additional confounding factors may limit the generalizability of our findings and encourage future research to address these gaps. In the meantime, we have revised the discussion section of our manuscript to highlight these limitations and to provide context for interpreting our results. Thank you once again for your constructive comments, and we hope that our revisions and explanations will address your concerns.
- The abstract should provide more thorough information about the genomic studies and their results; only JAK pathway is mentioned, without any additional explanation.
Response:Thank you for your valuable feedback on our abstract. We appreciate your concern regarding the thoroughness of the information provided about the genomic studies and their results. In response to your comments, we understand that the mention of the JAK-STAT signaling pathway in the abstract was indeed brief and lacked detailed explanation. We would like to clarify that our study indeed delved deeper into the genomic aspects, but due to space limitations in the abstract, we could only provide a brief overview. However, we recognize the importance of providing more information to better contextualize our findings. We have added some content in the abstract. Thank you again for your constructive comments.
- The genomic study is insufficiently developed. The limited number of genes analyzed and the reduced number of SNPs are insufficient for drawing any solid conclusions. Also, it is well known that epigenetic changes alter lipid and uric acid metabolism, and the Authors mention nothing about this.
Response: Thank you for your valuable feedback on our manuscript. We acknowledge the concerns raised regarding the genomic study and the lack of discussion on epigenetic changes.
Regarding the genomic study, we fully understand the limitations of the current analysis due to the limited number of genes and SNPs examined. However, it is important to note that the primary objective of our study was to investigate the relationship between the intake of solid block dairy products like cheese and serum uric acid levels, along with its potential physiological mechanisms. Despite the limitations, our results provide preliminary evidence supporting a negative association between cheese intake and serum uric acid levels, as well as a potential causal relationship inferred through two-sample Mendelian randomization analysis. We appreciate the suggestion to expand the genomic analysis in future studies. Indeed, we wish to conduct more comprehensive genetic studies with a larger sample size and a broader range of genetic markers to further elucidate the genetic underpinnings of this relationship. As for the epigenetic changes, we admitted that we did not address this aspect in our manuscript. We acknowledge that epigenetic modifications can play a significant role in altering lipid and uric acid metabolism. However, given the scope and focus of our current study, we did not include an analysis of epigenetic changes. We apologize for any oversight in this regard and appreciate the reviewer's guidance.
We are grateful for your constructive comments and suggestions. They will undoubtedly help us improve our research and provide more comprehensive insights into the relationship between dairy product intake and serum uric acid levels.Thank you again for your time and consideration.
- Did the Authors take any brothers/twins in the study? This should provide some interesting conclusions about the effects of diet in genetically close or similar individuals.
Response:Thank you very much for your question. However, twins were not involved in this study. Because we selected participants in the following way, and the probability of inclusion into twins though this sampling method was very low. That is, from 2016 to 2018, this project utilized a multi-stage stratified random sampling method to select 13 monitoring sites across Shandong provinces in China. The schools were chosen using a simple random sampling method, followed by the random selection of one class from each selected school and grade. Subsequently, 28 male and female students were randomly chosen based on their student ID numbers to participate in the survey. The brief sampling process is illustrated in this figure:
- The genomic analysis methodology is missing entirely; how was the DNA/RNA prepared, how was the sequencing performed, what controls were used etc.
Response:Thank you very much for your question. Due to space constraints, we previously abbreviated this section, but detailed introductions and steps have been supplemented in the "Bioinformatics Analysis" section of the method. Single nucleotide polymorphism (SNP) primarily refers to the polymorphism in DNA sequences caused by a single nucleotide variation at the genomic level, mainly arising from transitions or transversions of a single base. SNPs can be located in exons, introns, gene expression regulatory regions, and so forth, thereby exerting various effects on gene expression through these genetic variations. The fundamental approach of our analysis is to focus on the gene level from the SNP level, aiming to identify the functional effect genes corresponding to SNPs and then conduct functional enrichment analysis to infer potential physiological mechanisms between cheese intake and serum uric acid levels. Our perspective extends beyond the SNP level, as we believe identifying the genes influenced by SNPs is more meaningful. We utilized the FUMA website and R language for SNP mapping, followed by subsequent Gene Ontology (GO) and Protein-Protein Interaction (PPI) analysis of the mapped genes using the DAVID and STRING websites. These bioinformatics analysis methods do not involve experimental operations that require hands-on work, such as DNA/RNA preparation, sequencing, and other steps. These ideas and methods have been widely applied in existing related research (doi: 10.1016/j.jad.2024.03.098; doi: 10.1016/j.xcrm.2022.100805; doi: 10.1186/s12891-024-07186-7).
- Talking about controls, these are also missing entirely; similar investigations should have been performed on a group of subjects having a non-diary based nutrions, for comparison.
Response: Thank you for your thoughtful comments on our manuscript. We appreciate your concerns regarding the inclusion of a control group in our study. However, we would like to say that our study was a cross-sectional analysis based on data from the Chinese Children and Lactating Women Nutrition and Health Surveillance (CCLWNHS). In cross-sectional studies, the comparison is typically made between different exposure levels within the same population, rather than between separate control and exposed groups. The aim of our study was to investigate the associations between dairy product intake, particularly solid block dairy products such as cheese, and serum uric acid levels in a representative sample of Chinese children and adolescents. By using a standardized food frequency questionnaire and collecting comprehensive demographic, socioeconomic, and health behavior data, we were able to adjust for potential confounding factors and analyze the effects of dairy product intake on serum uric acid levels. While a longitudinal study or a randomized controlled trial would indeed provide stronger evidence for causality, such designs are not feasible with the existing CCLWNHS dataset. Instead, we focused on leveraging the rich information available within the dataset to generate hypotheses and insights into potential associations between dairy intake and serum uric acid levels. We agree that comparisons with a non-dairy-based nutrition group could provide additional perspectives, but such comparisons are not feasible with our current data. However, we have conducted sensitivity analyses, including stratified and interaction analyses, to ensure the robustness of our results across different subgroups. Furthermore, we have employed restricted cubic splines to analyze the dose-response relationship between cheese intake and serum uric acid levels, providing a more nuanced understanding of this association.
In conclusion, we understand the limitations of cross-sectional studies in providing definitive causality, so Mendelian Randomization was used to compensate for this deficiency. We believe that our analysis contributes valuable insights and hypotheses that can guide future research in this area. We are committed to pursuing further validation through more rigorous study designs in the future. Thank you once again for your constructive comments.
- The gene expression analysis, SNP analysis and especially the theoretical protein-protein interactions are highly speculative and do not provided reliable information in the absence of mandatory lab experiments.
Response: Thank you for your valuable feedback on our manuscript. We fully acknowledge that the bioinformatics analysis methods, including gene expression analysis, SNP analysis, and theoretical protein-protein interactions, are inherently speculative and may not provide definitive conclusions in the absence of mandatory laboratory experiments. These were our regrettable and inevitable limitation. However, we would like to emphasize that these methods serve as a crucial first step in generating hypotheses and providing insights into potential physiological mechanisms. The bioinformatics approaches employed in our study allow us to identify potential associations and trends that can guide future experimental research. By leveraging large-scale genomic data and advanced computational tools, we are able to screen for SNPs that may be functionally relevant and map them to genes that could be involved in the relationship between cheese intake and serum uric acid levels. This information, although speculative, provides a foundation for subsequent experimental validation. Furthermore, the results of our bioinformatics analysis can inform the design of animal studies by highlighting specific genes and pathways that warrant further investigation. By focusing on these targets, researchers can conduct more targeted and efficient experiments, potentially saving time and resources. In conclusion, while we recognize the limitations of bioinformatics analysis in providing definitive conclusions, we believe that our study contributes valuable insights and hypotheses that can guide future experimental research in this area. We are committed to pursuing further validation through laboratory experiments in the future. Thank you once again for your constructive comments.
Reviewer 3 Report
Comments and Suggestions for Authors
The study showed interesting findings, while some parts could be reconsidered. This is not a criticism but for a point of improvement of the paper.
1. Hyper(or hypo)uricemia includes a specific genetic background with a familiar history. Hyper(or hypo)uricemia with such genetic backgrounds differs clearly from that without in terms of clinical figures and dietary therapies. This should be detailed more. How about the prevalence of genetic uric acid disorders in your country? How was it approached and/or considered in the study? Were such family histories taken exactly in the study?
2. Does an individual cheese preference last from childhood to older ages? The dietary preference may change by dietary elements.
3. (How) do the parents’ preference affect the cheese preference of their children?
4. Cheese has various ingredients. Please introduce the representative kinds of elements and the proportion of constituents in your country and other countries.
5. Could the study indicate the threshold of Cheese intake for hyperuricemia?
6. In Methods, the participants were children. The way of informed consent should be detailed more (i.e., written consent?, the responsibility…).
7. In Methods, what was the principle for the selection of schools?
8. In Methods, the measurement methods including assay performance, reagent company, and device company for uric acid, glucose, and lipids could be detailed.
9. In Methods, why did the authors hear the alcohol consumption for children?
10. In Methods, the definitions of rural and urban areas should be described with any references.
11. In Methods, there were several URL citations. The accessed date could be added in the URL.
12. In Abstract and Results, were the beta-coefficients standardized? The coefficients could be expressed as standardized values.
13. In Table 1, 2 of ‘m2’ in the unit of BMI should be expressed in the superscript style.
14. In Table 3, why did the authors split/divide the age in 11? Please explain it.
15. The expression ‘Fig.’ and ‘Fig’ were mixed in Results.
16. Line 35; the end of the sentence can have some references.
17. Line 46; the end of the sentence can have some references.
18. Line 61; TSMR was here abbreviated. After that, the authors should not abbreviate it repeatedly.
19. Line 101, was the expression cross-sectional ‘study’?
20. Line 104; the term GLMs was here abbreviated. After that, the authors should not abbreviate it repeatedly.
21. Line 127; GO was here abbreviated. After that, the authors should not abbreviate it repeatedly.
22. Line 129; PPI was here abbreviated. After that, the authors should not abbreviate it repeatedly.
23. Line 135, 142, 143, and 146; SD can be full spelled out.
24. Line 223; beta < 0?
25. No space and a half-width space between the sentence and the reference No. were mixed (line 236 and 239).
Author Response
- Hyper (or hypo)uricemia includes a specific genetic background with a familiar history. Hyper(or hypo)uricemia with such genetic backgrounds differs clearly from that without in terms of clinical figures and dietary therapies. This should be detailed more. How about the prevalence of genetic uric acid disorders in your country? How was it approached and/or considered in the study? Were such family histories taken exactly in the study?
Response:Thank you for your thoughtful comments on our manuscript. Hyperuricemia, often associated with genetic predispositions, can be classified into primary (genetic) and secondary (acquired) forms. In families with a history of hyperuricemia, genetic factors like mutations of the URAT1 or ABCG2 genes can significantly impact uric acid metabolism, leading to higher serum uric acid levels compared to those without such genetic backgrounds (doi: 10.1186/s13075-021-02569-w). Clinically, patients with genetic forms may exhibit earlier onset, more severe manifestations, and a distinct response to dietary therapies, emphasizing the importance of personalized dietary management (doi:10.1053/j.ackd.2006.01.008). However, we were unable to find precise research data on the prevalence of genetic uric acid disorders in China, and relevant investigations were scarce. The ethics approval form explicitly stated that genetic analyses were not included in this study; therefore, we regret that we could not assess whether participants with hyperuricemia had any associated genetic backgrounds. Regarding family history, we only inquired whether participants' relatives had common hereditary diseases such as hypertension, diabetes, or asthma, and did not include hyperuricemia in the questionnaire. Unfortunately, we could not fully analyze genetic factor in this study, which was one of the limitations of this article.
- Does an individual cheese preference last from childhood to older ages? The dietary preference may change by dietary elements.
Response:Thank you very much for your question. An individual's cheese preference can be influenced by various factors throughout their life, and while some preferences established in childhood may persist, they are not fixed. Research indicates that early exposure to different flavors and textures can shape a child's taste preferences; however, these preferences can evolve due to several dietary elements and life experiences. As individuals age, their taste buds and sensory perceptions can change, leading to shifts in dietary preferences. Factors such as health considerations, dietary restrictions, cultural influences, and social contexts can significantly impact cheese choices. For instance, someone who preferred a specific type of cheese in childhood may later develop a preference for different varieties based on dietary health goals or lifestyle changes, such as adopting a vegetarian diet. Moreover, new experiences, such as travel or exposure to diverse cuisines, can introduce individuals to different cheese types, further influencing their preferences. Therefore, while childhood preferences may lay the groundwork, they are subject to change throughout an individual's life as they encounter new tastes and dietary influences (doi: 10.3390/nu9020107; doi: 10.3390/nu16030428; doi: 10.3390/nu13030983). Unfortunately, considering various factors such as budget, this project did not follow up in depth with each participant, and the results should be interesting if data on the participants' changes in cheese preferences were available over the next few years.
- (How) do the parents’ preference affect the cheese preference of their children?
Response:Thank you very much for your question. The influence of parents' preference on the cheese preference of their children is a multifaceted phenomenon. Research suggests that parental modeling, both explicit and implicit, plays a significant role in shaping children's food preferences. Explicitly, parents may introduce their children to specific types of cheese that they themselves enjoy, thereby fostering an early appreciation and preference for those varieties. Implicitly, parents' attitudes and behaviors around food, including their expressions of enjoyment or disgust towards certain foods, can subtly influence children's developing tastes. Furthermore, children often adopt the preferences and behaviors of their caregivers as a means of gaining approval and belonging. Thus, if parents express a strong preference for a particular type of cheese, children may develop a similar liking in order to align with their parents' tastes and gain social validation (doi: 10.3390/nu10060706; doi: 10.1111/j.1748-720X.2007.00111.x). We do believe that parents’ preference affect the cheese preference of their children. Family provide most of the food for children. And we hope this finding can increase cheese awareness for children and their parents.
- Cheese has various ingredients. Please introduce the representative kinds of elements and the proportion of constituents in your country and other countries.
Response:Thank you very much for your suggestion. We briefly presented the approximate ingredients of the cheese in the section of introduction. Cheese is made from various ingredients that contribute to its flavor, texture, and nutritional value. The primary ingredients in cheese include:
Milk: The main ingredient, which can come from various animals such as cows, goats, and sheep. The type of milk used significantly influences the cheese's flavor and texture.
Proportion: Typically, about 80-90% of the final cheese product consists of milk.
Rennet: An enzyme used to curdle the milk, separating it into curds and whey.
Proportion: Generally used in small amounts, around 0.05-0.1% of the total milk volume.
Bacterial Cultures: These are added to ferment lactose and develop flavor and texture.
Proportion: Usually, cultures make up about 0.5-1% of the total ingredients.
Salt: Enhances flavor and acts as a preservative.
Proportion: Typically, about 1-2% of the total weight of the cheese.
Flavorings and Additives: Herbs, spices, or other flavorings may be added.
Proportion: Varies widely depending on the type of cheese and can range from a few grams to several percent.
In China, cheese consumption has been increasing in recent years, with a growing variety of cheeses being produced, such as Pine Cheese (a type of soft cheese). The ingredients tend to follow similar proportions, focusing on cow's milk and local flavorings. In contrast, countries with a rich cheese-making tradition, like France and Italy, produce a diverse range of cheeses such as Camembert, Parmesan, and Mozzarella. These cheeses often incorporate unique local bacterial cultures and have specific aging processes, which affect the final product's flavor and texture.
5.Could the study indicate the threshold of Cheese intake for hyperuricemia?
Response: Thank you for your valuable comments on our study regarding the indication of a Cheese intake threshold for hyperuricemia. We appreciate the insightfulness of your query and have carefully considered it in the context of our research. Regarding the threshold for Cheese intake in relation to hyperuricemia, we acknowledge that determining such a threshold would require a comprehensive review of existing literature, input from experts in the field, and potentially large-scale studies to establish a statistically significant and clinically relevant threshold. Unfortunately, our current study, with its limited sample size and scope, does not provide sufficient data to support the derivation of a definitive threshold. Moreover, the potential for other adverse effects associated with excessive Cheese consumption necessitates a multifaceted assessment that considers various health parameters beyond serum uric acid levels. Our current dataset does not permit such an extensive evaluation, and therefore, we refrain from making arbitrary recommendations that could potentially mislead the readers.
While we recognize the importance of identifying a Cheese intake threshold for hyperuricemia, our study is not equipped to provide such a conclusion. We hope that future research, with larger sample sizes and more comprehensive methodologies, will address this gap in knowledge. Thank you once again for your constructive feedback, which has helped us to refine our understanding of the study limitations.
- In Methods, the participants were children. The way of informed consent should be detailed more (i.e., written consent?, the responsibility…).
Response:Thank you very much for your comments. We supplemented informed consent forms and the picture of ethical review documents for children aged 6-17 years. The informed consent form was signed by the participants, their guardian and the project staff to ensure that the interests of the involved population have been fully protected and that the subject may outweigh the possible risks.
- In Methods, what was the principle for the selection of schools?
Response: From 2016 to 2018, this project utilized a multi-stage stratified random sampling method to select 13 monitoring sites across Shandong provinces in China. The schools were chosen using a simple random sampling method, followed by the random selection of one class from each selected school and grade. Subsequently, 28 male and female students were randomly chosen based on their student ID numbers to participate in the survey. The brief sampling process is illustrated in this figure:

- In Methods, the measurement methods including assay performance, reagent company, and device company for uric acid, glucose, and lipids could be detailed.
Response:Thank you very much for your suggestions. The instrument,reagents and company used for the relevant biochemical index detection have been supplemented in thesection of methods.
- In Methods, why did the authors hear the alcohol consumption for children?
Response:Thank you very much for your comments. The information for the personal nutrition and health questionnaire for elementary school students was obtained by inquiring their parents. The questionnaire includes an assessment of alcohol consumption behaviors, with the question: "Has your child ever consumed alcohol, such as white liquor, yellow wine, beer, or wine? Please note that this does not refer to merely taking a sip." For middle school students, the personal nutrition and health questionnaire was self-administered. The question remained the same: "Have you ever consumed alcohol, such as white liquor, yellow wine, beer, or wine? Please note that this does not refer to merely taking a sip." If the participant answered "yes, within the last 30 days," they are then prompted to provide information on the frequency of alcohol consumption. All participant information was collected by a survey team that was trained and coordinated by the project. The existence of alcohol consumption behaviors among Chinese children and adolescents has been documented in other studies (doi:10.1186/s12887-024-04807-x; doi: 10.16835/j.cnki.1000-9817.2020.01.018); Alcohol intake is a confounder often included into consideration. Therefore, we designed questions related to alcohol consumption behaviors in the questionnaire.
- In Methods, the definitions of rural and urban areas should be described with any references.
Response:Thank you very much for your suggestions. Since August 1, 2008, the ‘Regulations on the Classification of Urban and Rural Areas’ (Guo Han [2008] No. 60) explicitly state that, based on China's administrative divisions, the classification is determined using the jurisdiction of resident committees and village committees recognized by civil affairs departments as classification objects, and actual construction as the basis for division. Thus, the country's territory is divided into urban and rural areas. In this context, urban areas include urban districts and town areas. Urban districts refer to the resident committees and other areas connected to the actual construction in the municipal districts and cities without district designations. Town areas refer to the resident committees and other areas connected to the actual construction in the county seat and other towns outside the urban districts. Meanwhile, rural areas refer to regions outside the urban areas defined in these regulations (2006, October 18. Regulations on the Classification of Urban and Rural Areas in Statistics. National Bureau of Statistics. https://www.stats.gov.cn/zs/tjws/tjbz/202301/t20230101_1903381.html) (Accessed October 27, 2024). We have added reference at the end of sentence. Please check the section of result.
- In Methods, there were several URL citations. The accessed date could be addedin the URL.
Response:Thank you very much for your suggestions. According to your advice, accessed date has been added. Please check the section of method.
- In Abstract and Results, were the beta-coefficients standardized? The coefficients could be expressed as standardized values.
Response:Thank you for your professional and meticulous advice. As you said, all beta coefficients are standardized partial regression coefficients. Thank you again for your professional advice.
- In Table 1, 2 of ‘m2’ in the unit of BMI should be expressed in the superscript style.
Response:We appreciate your scrupulous review. We have made modifications according to your comments. The 2 of ‘m2’ has been shown in the superscript style.
- In Table 3, why did the authors split/divide the age in 11? Please explain it.
Response:Thank you for your professional and meticulous advice. The age range of participants was between 6 and 18, and we observed that the mean age of them was 11.38, so we divided the age in 11 in table 3.
- The expression ‘Fig.’ and ‘Fig’ were mixed in Results.
Response:We appreciate your scrupulous review. We have made modifications according to your comments. Please check the section of results.
- Line 35; the end of the sentence can have some references.
Response:Thank you very much for your suggestion. We have made modifications according to your comments. We have added reference at the end of sentence. Please check the section of introduction.
- Line 46; the end of the sentence can have some references.
Response:Thank you very much for your suggestion. We have made modifications according to your comments. We have added reference at the end of sentence. Please check the section of introduction.
- Line 61; TSMR was here abbreviated. After that, the authors should not abbreviate it repeatedly.
Response:We appreciate your scrupulous review. We have made modifications according to your comments. Please check the section of results.
- Line 101, was the expression cross-sectional ‘study’?
Response:Thank you very much for your comments. We have changed the title to statistical analysis, and this statement was somewhat more accurate.
- Line 104; the term GLMs was here abbreviated. After that, the authors should not abbreviate it repeatedly.
Response:We appreciate your scrupulous review. We have made modifications according to your comments. Please check the section of results.
- Line 127; GO was here abbreviated. After that, the authors should not abbreviate it repeatedly.
Response:We appreciate your scrupulous review. We have made modifications according to your comments. Please check the section of results.
- Line 129; PPI was here abbreviated. After that, the authors should not abbreviate it repeatedly.
Response:We appreciate your scrupulous review. We have made modifications according to your comments. Please check the section of results.
- Line 135, 142, 143, and 146; SD can be full spelled out.
Response:Thank you very much for your suggestions. We have made modifications according to your comments. SD has been full spelled out. Please check the section of results.
- Line 223; beta < 0?
Response:Thank you for your professional advice and reminder. After adjusting for covariates, the intake of solid block dairy products, such as cheese, showed a consistently significant negative association with serum uric acid across all models. However, the statement of beta <0 was not accurate enough, so we have changed the beta to specific values.
- No space and a half-width space between the sentence and the reference No. were mixed (line 236 and 239).
Response:We appreciate your scrupulous review. We have made modifications according to your comments. The space between all the references and the sentences has been one.
Round 2
Reviewer 2 Report
Comments and Suggestions for Authors
During this second round of review, the authors have generally provided reasonable explanations for the limitations of their study and have made the necessary modifications in the text, to reflect the preliminary nature of their research. The other issues raised during the previous review were satisfactorily addressed however, due to the fact that some important variables cannot be acquired for this study, due to logistical reasons, and some confounding factors were not accounted for, I should change the title of the manuscript to reflect the fact that more research is still needed to draw some final solid conclusions. The title should read: "Effect of the intake of solid block dairy products like cheese on serum uric acid in Children:a preliminary mechanistic investigation".
Author Response
1. During this second round of review, the authors have generally provided reasonable explanations for the limitations of their study and have made the necessary modifications in the text, to reflect the preliminary nature of their research. The other issues raised during the previous review were satisfactorily addressed. However, due to the fact that some important variables cannot be acquired for this study, due to logistical reasons, and some confounding factors were not accounted for, I should change the title of the manuscript to reflect the fact that more research is still needed to draw some final solid conclusions. The title should read: "Effect of the intake of solid block dairy products like cheese on serum uric acid in Children:a preliminary mechanistic investigation".
Response: We appreciate your scrupulous review and meaningful suggestion for our manuscript. Thanks so much. This article title you suggested is more appropriate to our research content.
Reviewer 3 Report
Comments and Suggestions for Authors
The paper was being improved.
1. If beta coefficients were standardized, the coefficients appeared to be low. How do the readers accept the results?
2. The representative kinds of elements and the proportion of constituents in your country could be described more in the text.
3. For the informed consent, the word “written” could be added.
4. In the limitation, the comment on family history taken could be more added.
5. In the limitation, how (big/great or less) did no information medicine uses affect the results? What do the authors feel? The comment could be added.
6. In Methods, the measurement assay performance (coefficient of variation) was lacking.
7. In Methods, the principle to split/divide the age in 11 could be more described.
8. The places (City and Country) of companies of measurement assays could be added.
9. If possible, the comment on a Cheese intake threshold could be described more.
10. The expression ”2” in R2 may be described in a superscript.
Author Response
1. If beta coefficients were standardized, the coefficients appeared to be low. How do the readers accept the results?
Response: Thank you for bringing this point. You have raised a valid concern regarding the seemingly low beta coefficient (-0.182) observed in the relationship between cheese intake and serum uric acid levels in our study. I understand that this may initially seem to imply a modest effect size. The reduction in serum uric acid is a multifaceted process influenced by numerous factors. Cheese, as part of a balanced diet, may contribute to this reduction, but its individual effect appears modest in our study. This is consistent with the complexity of human nutrition and metabolism, where individual food items often exert subtle yet significant effects within the broader dietary context. Furthermore, the beta coefficient reflects the association between cheese intake and serum uric acid levels while holding other variables constant. In real-world scenarios, individuals' dietary habits, lifestyle factors, and genetic predispositions can all influence serum uric acid levels. Moreover, it is worth mentioning that even a modest reduction in serum uric acid can have clinically significant implications, particularly for individuals at risk of gout or hyperuricemia. Small changes in dietary intake, when maintained consistently over time, can lead to cumulative health benefits.
In conclusion, while the beta coefficient may suggest a relatively modest association between cheese intake and serum uric acid levels, this finding is consistent with the complexity of human nutrition. The modest effect does not diminish the potential clinical relevance of our findings, particularly when considering the broader dietary and lifestyle context in which cheese consumption occurs. Thank you again for your thoughtful comments.
2. The representative kinds of elements and the proportion of constituents in your country could be described more in the text.
Response: Thank you so much for your suggestion. The representative kinds of elements and the proportion of constituents of cheese has been added in line 54-56.
3. For the informed consent, the word “written” could be added.
Response: Thank you so much for your scrupulous review. We have revised them all in line 83, 385 and 387. Please check them.
4. In the limitation, the comment on family history taken could be more added.
Response: Thank you so much for your suggestion. The comment on family history taken has been added in line 345-348 in the limitation.
5. In the limitation, how (big/great or less) did no information medicine uses affect the results? What do the authors feel? The comment could be added.
Response: Thank you for raising this pertinent question regarding the impact of medication use on our results. Our study was specifically designed to investigate the correlation between cheese intake and serum uric acid levels. As such, we focused on dietary factors, with cheese being the primary variable of interest. The participates of our study are children, only those without underlying conditions or unaware of having any underlying diseases were included. The incidence of basic disease among them is very low, with little or no medicine use recorded. Moreover, the primary aim of our study was to identify the effect of cheese intake on serum uric acid, thereby minimizing potential confounding factors. Including medication use could have introduced complexity and uncertainty, as many medications known to affect uric acid levels (e.g., diuretics, allopurinol) are prescribed for a wide range of conditions unrelated to diet. Meanwhile, the sample size and scope of our study questionnaire did not allow for a comprehensive analysis of medication use. Accurate information on medication types, dosages, and durations would have been difficult to obtain and verify, potentially leading to bias or misleading conclusions. While medication use is undoubtedly an important factor in serum uric acid regulation, its exclusion from our analysis does not invalidate our findings. The observed negative correlation between cheese intake and serum uric acid levels remains a notable and worthy finding, particularly given the increasing focus on dietary interventions for managing uric acid levels.
In conclusion, while acknowledging the potential limitations of not considering medication use, we believe that our study provides valuable insights into the relationship between cheese intake and serum uric acid levels. Future research, with larger sample sizes and more comprehensive data collection, may further explore the role of medication use in this context. The comment has been added in line 350-353. Thank you once again for your constructive feedback.
6. In Methods, the measurement assay performance (coefficient of variation) was lacking.
Response: We appreciate your scrupulous review on our manuscript. The measurement assay performance (coefficient of variation), which was written as CV in manuscript, has been supplemented in line 90-96.
7. In Methods, the principle to split/divide the age in 11 could be more described.
Response: Thank you so much for your suggestion. We have described that 11 was based on the average age in line 123-124.
8. The places (City and Country) of companies of measurement assays could be added.
Response: Thank you so much for your suggestion. The information has been added in the section of method.
9. If possible, the comment on a Cheese intake threshold could be described more.
Response: Thank you so much for your suggestion. The limitation about the inability to derive exact thresholds has been added in line 353-357.
10. The expression ”2” in R2 may be described in a superscript.
Response: Thank you so much for your scrupulous review. We have revised them all in line 149 and 150. Please check the section of method.